# Transition in Sickle Cell Disease (SCD): A German Consensus Recommendation

**DOI:** 10.3390/jpm12071156

**Published:** 2022-07-17

**Authors:** Ferras Alashkar, Carmen Aramayo-Singelmann, Janine Böll, Annette Hoferer, Andrea Jarisch, Haytham Kamal, Lena Oevermann, Michaela Schwarz, Holger Cario

**Affiliations:** 1Department of Hematology and Stem Cell Transplantation, West German Cancer Center, University Hospital Essen, University of Duisburg-Essen, 45147 Essen, Germany; 2Department of Pediatrics III, University Children’s Hospital Essen, University of Duisburg-Essen, 45147 Essen, Germany; carmen.aramayo-singelmann@uk-essen.de; 3Department I of Internal Medicine, Center for Integrated Oncology Aachen Bonn Cologne Düsseldorf, University Hospital of Cologne, 50923 Cologne, Germany; janine.boell@uk-koeln.de; 4Hematology/Oncology Department for Adolescent Medicine, Robert-Bosch-Hospital, 70376 Stuttgart, Germany; anette.hoferer@rbk.de; 5Department for Children and Adolescents, Division for Stem Cell Transplantation and Immunology, University Hospital Frankfurt, 60590 Frankfurt am Main, Germany; andrea.jarisch@kgu.de; 6Joint Practice for Hematology and Oncology, 30625 Hannover, Germany; kamal.haytham@t-online.de; 7Department of Pediatric Oncology & Hematology, Charité University Medicine, 10117 Berlin, Germany; lena.oevermann@charite.de; 8Department of Hematology, Oncology, and Tumor Immunology, Charité University Medicine, 10117 Berlin, Germany; michaela.schwarz@charite.de; 9Department of Pediatrics and Adolescent Medicine & Center for Rare Hematopoietic Disorders and Immunodeficiencies (ZSHI Ulm), Ulm University Medical Center, 89069 Ulm, Germany; holger.cario@uniklinik-ulm.de

**Keywords:** sickle cell disease (SCD), transition, Germany, child blood non-malignant diseases, patient registries and standardization, ethics issues and patient-oriented perspectives

## Abstract

Sickle cell disease (SCD) is considered a rare disease in Germany. Due to the increasing prevalence, the acute and chronic morbidities associated with the disease and the sharp increase in the mortality rate of young adults, a need-based transition structure for patients with SCD in Germany is explicitly required. This is the first multicenter German consensus statement addressing the importance of implementing a standardized transition guideline that allows adolescents and young adults to safely transition from pediatric to adult care. Early identification of medical needs and intervention remains important in the context of chronic diseases. Effective measures can improve health care in general, as they lead to a reduction in disease and the consequential economic burden. It is noteworthy that improving structural barriers remains a key challenge even in highly developed countries such as Germany. Inclusion of these transition services for patients with SCD into the regular care of chronically ill adolescents and young adults should be ensured, as well as the coverage of costs associated with a structured transition process.

## 1. Introduction

Transition is the purposeful, planned transfer process of adolescents and young adults with chronic diseases from child-centered to adult-centered health care, with the goal of providing coordinated, continuous high-quality medical care without interruption. Transition encompasses the entire process of becoming an adult, and, in addition to enhancing autonomy and personal responsibility, it includes the effective use and provision of structures that enable the transition to adulthood for patients with chronic disorders [1].

To date, no uniform transition recommendations for patients with sickle cell disease (SCD) in Germany exist. However, the medical need for the implementation of a national structured and uniform transition process for patients with SCD is obvious considering the following facts:The prevalence of SCD in Germany has been increasing tremendously over the past years [2,3,4,5,6,7].The increase in the mortality rate of patients with SCD coincides with the period of transition to adulthood in Germany [8].Emergency care utilization among adolescents and young adults is increasing due to acute and/or progressive disease severity in this age group [9,10,11,12].These patients suffer from reduced quality of life (QoL) and increased psychological distress as compared with their unaffected peers and/or patients with other chronic disorders [13,14,15,16].

To address this need for guidance related to a transition structure for SCD patients, a network of pediatric and adult hematologists established an initiative to develop a standardized transition concept for patients affected by SCD in Germany.

## 2. Materials and Methods

The expert panel consisted of four pediatric and five adult hematologists representing either university medical centers or large community hospitals or working in private doctors’ offices in different parts of Germany.

The development of the consensus recommendations was based on the evidence-based guidelines for the transition from pediatric to adult care developed by the German Society for Transition Medicine (GSTM) (AWMF S3-guideline 186/001), which was published in April 2021 [1]. These guidelines contain overarching general recommendations covering all aspects of transition that should be considered in all patients, regardless of the underlying disorder. The aim of the current consensus process was the translation of those general recommendations into specific guidelines for the transition of patients with SCD. With such a consensus statement, the authors aim to improve the long-term situation of patients suffering from SCD and to lay the foundation for a need-based transition structure for SCD patients in Germany.

The process of the consensus statement development started in October 2021, with an initial online conference to create working groups, each drafting a statement concerning a subset of questions based on the general GSTM guidelines. Drafts were circulated and commented on through a Delphi-like process. Then, statements were discussed in two additional conferences, with another circulating in between. Finally, they were agreed upon by all participants.

## 3. Results and Discussion

### 3.1. Transition Policy and Adequate Planning Requires Education of Patients and Their Parents/Guardians with Identification of Individual Tasks

The transition from pediatric to adult care is a dynamic process rather than a cross-sectional event and, first and foremost, must be adapted to the adolescent’s individual developmental status. Adolescents with chronic diseases have specific needs compared with healthy peers. These needs must be considered in the decision-making process to ensure the best possible successful transition to adult care. In addition to individual development, family structures must be considered. Particular attention should be paid to SCD-associated complications that might have manifested in early childhood, which can further complicate the process of transition [17,18,19,20]. Individual roles with specific task assignments are summarized in Table 1.

In this regard, it is recommended to conduct age-adapted training sessions with the adolescents and their parents/guardians to discuss the specific aspects of the disease across different age groups (e.g., splenic palpation, proceedings in case of fever, vaso-occlusive crises (VOCs) or other complications, sufficient hydration, the necessity of vaccinations related to functional asplenia, etc.). Pre-existing knowledge should be refreshed, and, if necessary, new medical treatment options discussed. This will improve awareness of one’s disease, promote adherence and increase transition readiness and autonomy. Responsibility for disease management should gradually pass from the parents to the adolescent. Adolescents should become aware of the potential risks and consequences of not adhering to medical recommendations and therapies. When reaching the legal age of maturity, all adolescents should know important contacts outside of pediatric care to turn to in the event of acute complications (e.g., fever, VOCs, etc.) [12].
**Core Statements**The transition process should always be considered in the context of the family structure and must include the personal needs and developmental requirements of the family (e.g., navigating health insurance, teenage pregnancy and geographical changes due to college and/or employment).The patient (depending on age and maturity) and their parents/guardians are full partners and should be involved in the entire treatment and transition process.From the onset of puberty at the latest, adolescents with SCD should be encouraged to develop essential skills to manage their disease independently. The potential risks and consequences of not following recommendations/therapies should be known.Patients and parents/caregivers should have a good understanding of the disease. They should have learned how to recognize and manage signs, symptoms and associated complications in patient and family training sessions. Training sessions need to be repeated at regular intervals to discuss the specifics of the different ages, repeat what is known and discuss new medical developments as appropriate.In case of presentation in an outpatient clinic where the patient is unknown, he/she needs an emergency card in which the diagnosis, useful initial measures and emergency interventions are noted.

### 3.2. Structural Factors and Requirements Need to Be Established and Further Improved to Ensure Interdisciplinary Care

Structural factors (i.e., lack of qualified or networking care structures close to home or lack of knowledge on their availability) that might complicate the outcome of transition should not play an important role. However, they continuously represent a significant problem and a barrier to successful transition throughout Germany. For professional legal reasons, further treatment by pediatricians is generally not permitted after the age of 21, in some places even after the age of 18. The only exceptions are vaccinations and therapies in the context of emergencies. However, this conflicts with the general recommendations of the GSTM, as the timing of the transition to adult care should be adapted to the specifics of the disease and the individual patient and should not be rigidly tied to the age of maturity (18 years). Notably, in the case of inadequate treatment options by specialists, treatment can be provided by all pediatricians in accordance with the “pseudo-fee schedule item 99155” [21]. However, this should be outside the scope of statutory standard care, even though it can be provided in individual cases as a bridging measure in order not to jeopardize transition in patients with SCD.

### 3.3. Establishment of a Structured Transition Process and Assessment of Transition Readiness

The pediatric hematologist caring for the patient is responsible for determining the appropriate time for initiating the transition process. He develops an individual transition plan in which specific measures are precisely defined and scheduled, and he identifies potential challenges threatening the outcome of transition in the long term. The adolescent, as well as parents and caregivers, should be offered psychosocial support in advance. The involvement of self-help and patient advocacy groups additionally contributes to minimizing barriers. Therefore, adolescents and their parents/guardians must be actively informed about the availability of such organizations. Data have shown that personal involvement in self-help and patient advocacy groups, through sharing experiences among peers, has a positive impact on the self-determined management of chronic diseases [22,23,24].
**Core Statements**The transition process is led by the pediatrician in charge, who is responsible for providing comprehensive medical care in an interdisciplinary network.Every patient with SCD, as well as their parents/guardians, should be offered the option of psychological support.Adolescents or their parents/guardians should be actively informed about the availability of specific self-help and patient organizations.Patients should be informed of the availability of websites and mobile phone services (e.g., apps) at regular intervals to improve adherence and keep appointments. 

Transition readiness is reached when all decisions are made and actions are taken to build the capacity of the individual adolescent, and that of others involved in the patient’s environment, to prepare for, begin, continue, and finish the process of transition.

To assess transition readiness, the panel recommends the use of a standardized questionnaire, which can be used repeatedly to assess the personal capability of each adolescent individually [25,26,27,28,29,30,31,32].

It should be mentioned that despite the accessibility of continuous medical care in Germany, adolescents with chronic disorders often cannot benefit from specific therapy options due to individual psychoemotional deficits [33,34]. Therefore, an initial screening, including neurophysiological testing, should be carried out for every child with SCD before starting school, at the time of any change in school and again when the child reaches the age of 16. Of course, any corresponding medical abnormality would indicate such intervention as well. Thereby, medical caregivers from disciplines involved in comprehensive care are enabled to intervene in a timely manner (e.g., in case of adherence difficulties due to a psychological overload of the adolescent). In addition, such regular examinations will allow the early detection of disease-specific morbidities [12,35]. For patients with cognitive impairment, the involvement of parents and caregivers is mandatory.

Close contact with the general pediatrician and future general practitioner is demanded and essential. Specialists should be consulted at an early stage, taking already manifest disease-associated morbidities into account. They should also be known to the adolescent and their parents. Cooperation with a tertiary referral hospital in the field of adult medicine with expertise in SCD is demanded, as well as adherence to the guideline of the Joint Federal Committee on Quality Assurance Measures for the Inpatient Care of Children and Adolescents with Haemato-oncological Diseases [36].
**Core Statements**The patient’s readiness for transition should be assessed by means of a structured questionnaire, which might also be presented to the parents/guardians (form provided as part of the transition initiative). Neurophysiological testing should be carried out for every child with SCD before starting school, at the time of any change in school and again when the child reaches the age of 16. Of course, any corresponding medical abnormality would indicate such intervention as well.A joint exchange with the ambulant pediatrician, the future general practitioner as well as specialists should be guaranteed.

### 3.4. Transfer of Care 

Once the adolescent has reached transition readiness, the adolescent and parents/guardians—in agreement with the patient—should be assigned an adult medical professional with expertise in SCD [18].

Once an appropriate adult physician with the necessary expertise has been identified, the adult physician should be informed, ideally three to six months prior to the first joint meeting. As part of a comprehensive summary, the adult physician and general practitioner should be informed about the clinical course, including disease-specific complications and any current or planned upcoming therapies, to allow for a well-ordered medical transition. It is suggested to schedule an initial joint meeting in the presence of the parents/guardians, possibly with the involvement of social services. This first meeting should take place before the young adult reaches the age of maturity. Ideally, the pediatric expert would participate in this first visit. Optionally, this could also be conducted in the context of a virtual meeting. Depending on the needs, but especially in the case of emerging challenges that might endanger a successful transition outcome, the adolescent and the parents/guardians should be offered the opportunity for an additional joint meeting. For legal and other reasons, the patient’s consent for parental involvement at that time is required. This not only allows the adolescent the opportunity to build trust, but it also allows the opportunity for timely changes or the clarification of any misunderstandings that may exist from the patient’s point of view.
**Core Statements**Continuous specialized medical care throughout the vulnerable life phase of adolescence into young adulthood is crucial and must be ensured.The initial meeting with the adult physician should take place in the presence of the parents/guardians, and if logistically possible, be accompanied by the pediatrician.A structured transition letter for information transfer from pediatric to adult hematology is mandatory to ensure the adequate transfer of information and potential therapeutic options (form provided as part of the transition initiative) [31].

### 3.5. Completion of Transition and Opportunity of Follow-Up and Escalate when Needed

An option for follow-up by the pediatrician, at least within the first year after the transition has taken place, is recommended. Another measure to monitor transition is the inclusion of patients in the “Sickle Cell Disease” registry, supported by the Gesellschaft für Pädiatrische Onkologie und Hämatologie (GPOH). Based on the registry, follow-up complications will also be recorded as part of the annual status survey, thus offering the opportunity to measure the effectiveness of the transition [37]. It should be clear to the young adult, and also to the adult physician, that a transition back to pediatric care may be possible, at least in the interim, if serious problems arise that might jeopardize the success of the transition process (“pseudo-fee schedule item 99155”) [21].

In the context of adult care, young adults and their families/guardians should continuously be offered regular psychosocial support. It is well known that leaving a familiar environment presents an emotional stress factor. The goal is to improve care for these chronically ill young adults, strengthen compliance and ensure integration into the adult world, including medical care [38,39].
**Core Statements**Always give the patient the opportunity to transiently return to his pediatric specialist if this is desired (depending on local opportunities).In the first year after the transition, provide feedback on patient status to pediatric colleagues.

## 4. Conclusions

SCD is a multifaceted, multisystem disorder associated with acute and chronic complications due to complex pathophysiological mechanisms that can affect patients physically, psychologically and mentally. The increased morbidity and mortality as compared with unaffected peers, as well as the rarity of the disease in Germany, together with structural requirements, lead to a particular need for national guidance on a standardized process and structure for the transition of adolescents and young adults with SCD in Germany [3].

In addition to the national newborn screening (NBS) for SCD, which was introduced in October 2021, and guarantees the early identification of patients, the development and implementation of such a structured transition will further improve medical care, lead to a better quality of life and even increase chances of survival in the future, based on the prevention or at least early diagnosis and treatment of foreseeable chronic organ changes [40,41]. In addition, such a standardized transition process may be of significant health and economic relevance considering the increased need for acute medical care, especially in young adulthood, as well as the demanding treatment of chronic organ complications [42,43,44]. Therefore, the inclusion of these transition services in the regular care of chronically ill adolescents and young adults suffering from SCD and the coverage of financial requirements by health insurance companies are very important and required.

## Figures and Tables

**Table 1 jpm-12-01156-t001:** Individual task assignment.

**The Pediatrician:**
Determine the right time for transition (the age of 18 is not a fixed point).
Offer appropriate preparatory discussions and training sessions for the patient and their parents/caregivers at an early stage—around the age of 16 years (possibly as early as the age of 14 years).
Promote independence of the young person in managing his/her disease (e.g., adherence and independent scheduling of appointments, taking medication, etc.).
Strengthen the basis of trust in the patient’s future doctor/caregiver.
Establish early contact with future physician/caregiver, no later than 3-6 months prior to transition interview (exact timing depends on individual circumstances of patient and center).
Organize joint discussions with other disciplines (e.g., psychosocial service).
Provide a detailed, informative final letter (comprehensive summary) for future physicians/caregivers.
**The future medical adult caregiver:**
Obtain knowledge of the patient’s history, course, and current clinical status.
Provide early feedback to the pediatrician (tertiary hospital) on questions about treatment plans, adherence, patient’s psychosocial status, etc.
Prepare an individual therapy plan (previous therapy concepts should not be changed abruptly).
From the first contact with the patient, establish a basis of trust (a fixed contact person is required!).
Always give the patient the opportunity to return to pediatrics if this is desired, taking the network into account.
In the first year after the transition, offer feedback on patient status to pediatric colleagues.
**Patient:**
Be willing to participate in a scheduled transition process.
Stay aware of appointments in preparation for transition.
Be willing to strive for one’s own independence.
Bring life plans to the transition discussions to help manage the challenge posed by the illness.
Be willing to build a foundation of trust with new physicians/caregivers.
**Family/guardians:**
Build confidence in the child’s independence (e.g., taking medications, keeping appointments, etc.).
Build trust with the new doctors/caregivers.
Provide emotional support, especially in the first few months after the transition is complete.
Provide feedback to the pediatrician and/or current medical caregiver when difficulties occur.
**Psychosocial service:**
Participate in discussions with the patient and parents/guardians, if possible.
Perform sociolegal consultation (if not already conducted).
Provide advice on special reimbursement options.
Exchange with psychosocial services.
Support the patient and family/guardians throughout the entire transition process and thereafter.
Support life plans (e.g., schooling, education despite illness).

## Data Availability

Publicly available datasets were analyzed in this study. These data can be found here: https://www.sichelzellkrankheit.info/transition/ (accessed on 28 May 2022).

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
