# Peer review of "Transition in Sickle Cell Disease (SCD): A German Consensus Recommendation"

_jpm, 2022, doi:10.3390/jpm12071156_

Round 1

Reviewer 1 Report

In this article, the authors reconsidered the protocols for the transition of SCD patients from child-centered to adult-centered health care in Germany and offered recommendations for improvement.

There are some major issues with the manuscript;

1-      Improved informational content is needed in the manuscript.

2-      The authors did not provide specific and clear criteria (such as clinical or laboratory findings) for the time of transition. They only suggest that the age of transition should not be fixed.

3-      Comparison of their recommendations with the latest guidelines is recommended. (AWMF S3-guideline 186/001).

4-      The reasons for their recommendations were not appropriately discussed.

Author Response

Dear Reviewer,

thank you very much for the opportunity to revise our manuscript. Please find below our responses, addressing point by point the important issues raised.  

Improved informational content is needed in the manuscript.

Answer: We regret that you recommend a further improvement, as we believe, that we have addressed the key issues that require a structured transition process for patients living in Germany. It should also be taken into account that our recommendation - with participation of both, pediatric and adult physicians - is the first uniform recommendation in Germany, with the aim to improve a hitherto inadequate care structure for these often severely ill patients. 

The authors did not provide specific and clear criteria (such as clinical or laboratory findings) for the time of transition. They only suggest that the age of transition should not be fixed.

Answer: In our opinion, there are no specific laboratory or clinical findings as a prerequisite for transition. Of course, prior to and at the time of transition, a thorough clinical and laboratory assessment should be performed by the pediatric hematologist according to current guidelines to ensure the best possible information about the patient's current status to the colleagues providing future medical care. 
Therefore, we had paid special attention to the paragraph "Establishment of a structured transition process and assessment of transition readiness" in the manuscript, as this is one of the most important aspects of a successful transition, among many others (e.g., patient education and close collaboration between pediatric and adult physicians).  

Comparison of their recommendations with the latest guidelines is recommended. (AWMF S3-guideline 186/001).

Answer: As outlined in the Materials and Methods chapter, we have been guided by the aforementioned S3 guideline in developing this consensus recommendation. We have used the recommendations given there as an overarching practical recommendation tailored to the care of SCD patients. However, this recommendation is a general recommendation for chronically ill adolescents and young adults. Therefore a direct comparison might not be meaningful with regard to our recommendations.

The reasons for their recommendations were not appropriately discussed.

Answer: We had deliberately kept the discussion in the manuscript short, as we had already stated in the introduction the essential reasons for our recommendations. The final discussion or summary should therefore only refer to an outlook on the possible positive benefit of an implementation of our recommendations in conjunction with the newly available newborn screening in Germany. 

Finally, we would thank you for your constructive and important input, and we hope we were able to address all the issues raised.

Reviewer 2 Report

This is a very well written and very useful recommendation paper for effective transition of care from pediatric to adult hematologists for patients with sickle cell disease. The authors provide the gap in the field, the method of development of consensus guidelines, and then succinctly give the core statements. Here are some suggestions to improve the utility of such a consensus statement:

- Provide more reference for transition readiness assessment tools such as Nazareth et al (2018), Abel et al (2015), Perry et al (2017), Sawicki et al. (2011), Sobota et al (2014), and the ASH transition tool kit. 

- The authors should comment on transition benchmarks and how to monitor effectiveness of transition.

- Challenges such as education regarding navigating health insurance, teenage pregnancy, geographical changes due to college or employment need to be addressed.

Author Response

Dear Reviewer,

thank you very much for the opportunity to revise our manuscript. Please find below our responses, addressing point by point the important issues raised.

Provide more reference for transition readiness assessment tools such as Nazareth et al (2018), Abel et al (2015), Perry et al (2017), Sawicki et al. (2011), Sobota et al (2014), and the ASH transition tool kit. 

Answer: We used the ASH transition kit as one source to develop our readiness assessment form which can be downloaded at https://www.sichelzellkrankheit.info/transition/ (see also data availability statement). In addition, we used pre-existing questionnaires developed by the “Berliner TransitionsProgramm (BTP)” in collaboration with the scientific societies GPOH and DGHO. These important references, including the suggested references were added.  

The authors should comment on transition benchmarks and how to monitor effectiveness of transition.

Answer: Thank you very much for your comment. We have added this important point in the section "Completion of transition and opportunity of follow-up and escalate when needed" (Another measure to monitor transition is the inclusion of patients in registry "Sickle Cell Disease" supported by the Gesellschaft für Pädiatrische Onkologie und Hämatologie (GPOH). 

Challenges such as education regarding navigating health insurance, teenage pregnancy, geographical changes due to college or employment need to be addressed.

Answer: Thank you very much for your comment. We absolutely agree that all these points must be part of the educational program (pages 3 to 5). These specific and very important aspects have been included in the core statements. 

Finally, we would thank your for your constructive and important input. We hope,  we were able to address all the issues raised.

Round 2

Reviewer 1 Report

unfortunately, none of my comments has been responded properly and the revisions that I have asked has not been performed in the manuscript.